Movement patterns of two reintegrated African elephant (Loxodonta africana) herds: transitioning from captivity to free-living

http://orcid.org/0000-0001-6945-2409 Roos Tenisha 1 tenisha@elephantreintegrationtrust.co.za
Purdon Andrew 2
Boult Victoria 3
http://orcid.org/0000-0002-5899-3370 Delsink Audrey 4
http://orcid.org/0009-0001-5054-3252 Mitchell Brett 1
http://orcid.org/0000-0002-8942-4779 Kilian Petrus Johannes 1 5
1 Elephant Reintegration Trust , Port Alfred, Eastern Cape , South Africa
2 M.A.P Scientific Services , Pretoria, Gauteng , South Africa
3 Department of Meteorology, University of Reading , Reading , United Kingdom
4 Humane Society International-Africa , Cape Town , South Africa
5 !Khamab Kalahari Reserve , Tosca, North West , South Africa
Manjarrez Javier
Electronic publication date: 2024 Jun 6
Publication date: 2024
Volume: 12
Electronic Location ID: e17535
Received 2023 Dec 22; Accepted 2024 May 17
Copyright: © 2024 Roos et al.
Copyright year: 2024
Copyright holder: Roos et al.
License: This is an open access article distributed under the terms of the Creative Commons Attribution License, which permits unrestricted use, distribution, reproduction and adaptation in any medium and for any purpose provided that it is properly attributed. For attribution, the original author(s), title, publication source (PeerJ) and either DOI or URL of the article must be cited.
License URL: https://creativecommons.org/licenses/by/4.0/

Keywords: African elephant, Reintegration, Captive elephants, Movement patterns, Welfare

Funding: The Elephant Reintegration Trust (Tenisha Roos and Brett Mitchell) The Humane Society International-Africa NERC Knowledge Exchange Fellowship NE/V018841/1 The Elephant Reintegration Trust (Tenisha Roos and Brett Mitchell) received a grant from Humane Society International-Africa to produce the manuscript and cover the open-access cost. The Humane Society International-Africa also granted access to the collar data obtained from the collar they supplied as part of the immunocontraception programme on !Khamab Kalahari Reserve. Victoria L. Boult was supported by a NERC Knowledge Exchange Fellowship (NE/V018841/1). The funders had no role in study design, data collection and analysis, decision to publish, or preparation of the manuscript.

==============================
With the escalating challenges in captive elephant management, the study of elephant reintegration emerges as a pivotal area of research, primarily addressing the enhancement of animal welfare. The term ‘reintegration’ refers to the process of rehabilitating captive elephants to a natural system, allowing them to roam freely without intensive human intervention. There is a relative paucity of research addressing the behavioural adaptations post-reintegration, despite reintegration of over 20 elephants across various fenced reserves in South Africa. Our study centres on two distinct herds of reintegrated African elephants, monitoring their movement patterns in two South African reserves over a 57-month period post-release. The primary goal of the study was to establish whether the flexibility and adaptability of movement behaviour of reintegrated elephants can be considered as one of the indicators of determining the success of such an operation. The second aim of our study was to investigate if the reintegrated elephants demonstrated an adaptability to their environment through their hourly, daily, and seasonal ranging patterns after a period of free roaming that exceeded 4 years. Our findings indicated that reintegrated elephants, much like their wild counterparts (movement based on literature), displayed notable seasonal and diurnal variations in key movement parameters, such as utilisation distribution areas and reserve utilization. These patterns changed over time, reflecting an adaptive shift in movement patterns after several years of free roaming. Notably, the trajectory of changes in movement parameters varied between herds, indicating unique adaptation responses, likely resulting from differences in the reintegration process (familiarity of reserve, season of release, presence of wild elephants). Although our study is constrained by the limited number of reintegrated herds available for analysis, it underscores the potential of captive elephants to successfully adapt to a free-living environment, emphasising the promising implications of reintegration initiatives.

Introduction

In the face of escalating complexities in managing captive elephants (Loxodonta africana) and growing concerns regarding animal welfare (Rees, 2021), alternative methods to eliminate these welfare considerations are currently under investigation. Captive elephants include those who have been bred in captivity or those who have been wild caught to be kept in commercial exhibition facilities for the use of elephant-back-safaris, human-elephant interactions, zoos, safari parks and circuses (Department of Environmental Affairs and Tourism (DEAT), 2008) (this does not include orphaned elephants that refer to wild elephants that have lost their mother due to natural causes such as death). Captive facilities are often ill-equipped to manage unwanted behaviours such as aggression and frustration, and to mitigate risk, keepers may stop handling animals or restrain them further (Brando & Norman, 2023), increasing the lack of stimulation. This not only fails to solve the problem but can potentially lead to dire consequences such as human injury or fatalities (Wentzel & Hay, 2015). Factors such as restraint and decreased handling exacerbate the frequency of these unwanted behaviours (Jachowski, Slotow & Millspaugh, 2012; Szott, Pretorius & Koyama, 2019; Szott et al., 2020; Garai et al., 2022). This could lead to an elephant being deemed as unmanageable due to it either killing humans or conspecifics (Shaffer et al., 2019). Some facilities either sell or give away animals to other facilities or zoos when there is no longer a commercial value in keeping them. Despite the public concerns surrounding the wellbeing of captive elephants (Rees, 2022), alternative viewpoints suggest the availability of more cost-effective and replicable methods for improving captive elephant welfare instead of reintegration into a wild system (Phalke, 2020; Suter, 2020). Some researchers have argued that elephant reintegration could offer a practical solution for mitigating welfare challenges associated with captive elephants (Baker & Winkler, 2020) by rewilding them (Carter & Kagan, 2010) and reducing their numbers in captivity.

The study of elephant reintegration is a pivotal and somewhat contentious area of research. Elephant reintegration involves acclimatising captive elephants for a life in the wild, enabling them to freely roam without intense human intervention (Baker & Winkler, 2020). This practice has seen over 20 captive African elephants reintegrated into various fenced reserves in South Africa over the years (Table SA1). Although reintegration is not new in southern Africa, studies of both Asian and African elephants (Ashraf et al., 2005; Angkavanish & Thitaram, 2012; Evans, Moore & Harris, 2013a, 2013b; Perera et al., 2018; Goldenberg et al., 2021; Pretorius, Eggeling & Ganswindt, 2023) analysing the behavioural adaptations of elephants post-reintegration remain scarce and detractors still question the credibility and viability of this practice.

South Africa currently holds 90–95 captive elephants (Mitchell, 2019; Wentzel & Hay, 2015), which is unlikely to increase as the ‘National Norms and Standards for the Management of Elephants in South Africa’ no longer allows the capture of wild elephants to be kept in controlled environments (Department of Environmental Affairs and Tourism (DEAT), 2008). This does not include genuine orphaned animals. The South African conservation landscape differs from most others in Africa and consists of multiple fenced reserves varying in size and elephant numbers (Pretorius, Garaï & Bates, 2019), providing the opportunity for future reintegrations onto secure, wild managed systems.

Elephants possess a remarkable ability to discriminate across chemical, auditory, and visual sensory modalities, which allows them to communicate and increase their knowledge of unknown environments (Rasmussen & Schulte, 1998; McComb et al., 2000; Shoshani, Kupsky & Marchant, 2006; Byrne, Bates & Moss, 2009). This characteristic is acknowledged by Mumby & Plotnik (2018), who suggest that elephants exhibit immense potential for successful rehabilitation and release in a natural system. However, despite their inherent adaptability, doubts linger about the ability of previously captive elephants to acclimate to unfamiliar environments (Jensen & Tweedy-Holmes, 2007; Thulin & Röcklinsberg, 2020). Sceptics fear that released animals may not experience improved living conditions because of starvation, predation, disease, or hunting/euthanasia due to crop raiding or continued human association and interactions (Evans, Moore & Harris, 2013a) or continued human association and interactions that may be present in these unfamiliar fenced or open areas (Thulin & Röcklinsberg, 2020).

However, several elephant reintegration projects have been undertaken and documented (Ashraf et al., 2005; Angkavanish & Thitaram, 2012; Evans, Moore & Harris, 2013a, 2013b; Perera et al., 2018; Goldenberg et al., 2021; Pretorius, Eggeling & Ganswindt, 2023). These studies evaluated reintegration success by investigating various factors such as measures of health (dung (faecal glucocorticoid metabolites (fGCMs)), and body condition score), behaviour (feeding, drinking, moving, grooming, etc.,), ability to form social bonds with wild elephants, ranging patterns, breeding success, and the impact on the environment and surrounding human communities. However, there is a notable lack of studies focusing on the above-mentioned long-term success metrics such as longevity, breeding success (Evans, Moore & Harris, 2013a), social- and demographic learning (Goldenberg et al., 2019) of reintegrated elephants (Baker & Winkler, 2020).

The success of reintegration operations hinges on several factors. Age and social structures reflecting natural structures (B. Mitchell, 2015, personal communication), including at least the first and second family tier units (Wittemyer et al., 2007; Pretorius, Garaï & Bates, 2019), can significantly bolster the resilience of translocated animals (Goldenberg et al., 2019, 2021). To the contrary, Evans, Moore & Harris (2013b) demonstrated that reintegration can indeed be successful for individual elephants after monitoring a reintegrated African elephant cow for over 5 years post-release. Even though this elephant never integrated with a wild herd, she, along with another reintegrated female from the same initial group but a different reintegration program, established their own herd and successfully reproduced (Evans, Moore & Harris, 2013b).The age of the animal (Kleiman et al., 1991; Custance, Whiten & Fredman, 1999; Woodroffe & Ginsberg, 1999) also plays a pivotal role in the success of such an operation. In some species, adolescence is the optimal life stage for reintegration due to the enhanced behavioural flexibility and adaptability exhibited during this phase (Kleiman et al., 1991; Custance, Whiten & Fredman, 1999; Woodroffe & Ginsberg, 1999). However, reintegrating elephants without access to older, more experienced herd members might give rise to challenges such as impaired food acquisition, reduced social networking abilities, higher mortality rates, increased frequency of visits to the release site, and other behavioural issues (Thornton & Clutton-Brock, 2011; Garaï, Boult & Zitzer, 2023). Unnatural bull hierarchies (lacking the presence of different age groups, especially older bulls (Allen et al., 2020) resulting in unusual elephant musth patterns, have been demonstrated to contribute towards behavioural abnormalities. These include white rhinoceros (Ceratotherium simum) fatalities during elephant and rhino interactions (Slotow & Van Dyk, 2001). Furthermore, the method (soft- or hard release) of reintegration (Baker & Winkler, 2020), the captive history of the elephant, and the characteristics of the release site (Phalke, 2020) are important factors to take into account to ensure the success of a reintegration.

Parameters such as home range sizes, utilisation distribution, and daily displacements can provide valuable information about an elephant’s behavioural flexibility and adaptability to environmental changes (Benhamou, 2004; Wato et al., 2018), including seasonal variations in resource availability and extreme weather events (Wato et al., 2018). The parameters mentioned above will aid in investigating their foraging behaviour and seasonal variations in ranging behaviour (Young, Ferreira & Van Aarde, 2009; Wato et al., 2018).

Since the mid-1980s, South Africa has seen multiple privately- or state-owned small to medium-sized fenced reserves being established (Pretorius, Garaï & Bates, 2019), with the main goal of reintroducing various wildlife species, including the elephant (Duffy et al., 2002). As mentioned previously, behavioural abnormalities have been documented in reserves where elephants reside within incomplete social structures (Allen et al., 2020; Garaï, Boult & Zitzer, 2023). These unnatural behavioural occurrences are often a consequence of limited space available to host complete social structures required for normal interactions with conspecifics (Pretorius, Garaï & Bates, 2019), the introduction of cull orphans originally translocated from KNP (Kruger National Park) in the early 90’s (Carruthers et al., 2008), as well as conflicts with other species within the fenced system (Slotow & Van Dyk, 2001). Artificial boundaries could have a confounding effect on elephants as they isolate populations from one another as well as reduce their available space to exhibit seasonal movement patterns. If these fenced systems lack adequate space for natural movement, elephant movement patterns can become localised and inevitability reduction of available food sources for other species may occur (Landman, Schoeman & Kerley, 2013)

The movement patterns of wild elephants in open systems can vary significantly based on several factors such as the gender of the elephant (Moss, 1996; Whitehouse & Schoeman, 2003; Shadrack et al., 2017), rainfall season (Chamaillé-Jammes, Valeix & Fritz, 2007; Smit, Grant & Devereux, 2007; Wittemyer et al., 2007; Young, Ferreira & Van Aarde, 2009), diurnal cycle (Loarie, Aarde & Pimm, 2009), the presence of predators (Bleicher, 2017), topography (Duffy et al., 2011; Chibeya et al., 2021), elephant demography (Bates et al., 2008), and the number of artificial water points and electrified fences (Loarie, Aarde & Pimm, 2009). More specifically, the home range size of wild elephants varies across seasons and tends to be larger during the wet season (Leggett, 2006; Young, Ferreira & Van Aarde, 2009; Purdon & Van Aarde, 2017). This phenomenon is attributed to elephants’ decreased dependency on water during the wet season, allowing them to range further during this season (Chamaillé-Jammes, Valeix & Fritz, 2007; Smit, Grant & Devereux, 2007; Wittemyer et al., 2007; Young, Ferreira & Van Aarde, 2009). Wild elephants’ movement paths tend to be denser during the dry season, indicating that they visit the same areas more than in the wet season (Loarie, Aarde & Pimm, 2009; Wato et al., 2018). The daily displacement of wild elephants tends to be short during the middle of the day, and larger at dusk and dawn (Loarie, Aarde & Pimm, 2009). The presence and distribution of artificial water holes influences elephant movement patterns, by increasing the dry season range available to elephants as well as reducing the effect of the season (Viljoen, 1989; de Beer & van Aarde, 2008; Loarie, Aarde & Pimm, 2009; Purdon & Van Aarde, 2017).

In South Africa, reserves are fenced off and electrified forming physical boundaries that block elephants from roaming in the wet season, resulting in more concentrated movement near the fence (Loarie, Aarde & Pimm, 2009). Wild elephants’ ranges may decrease, and the same sites will be visited regularly during the wet season in the presence of fences (Loarie, Aarde & Pimm, 2009). The topographic characteristics of fenced- and open systems is another crucial driver of elephant movement patterns (Duffy et al., 2011; Evans et al., 2020; Hines et al., 2023). A study focussing on habitat connectivity identified three environmental factors (Normalized Difference Vegetation Index, elevation and landcover) that had a significant influence on the distribution of elephants during the wet season in Sioma (Zambia) landscape (Chibeya et al., 2021). Spatial elevation and gradient affect the dispersal and persistence of plants (West et al., 2016), which indirectly determine whether and how often elephants move in certain areas or not (Duffy et al., 2011). Elephants will utilise certain habitat systems such as riverine thickets that provides different resources (water, forage, and shade) (Shannon et al., 2006). In addition to elephants’ spatial acuity when assessing water sources (Chamaillé-Jammes et al., 2013), they exhibit the ability to remember the spatial locations of other elephants in relation to themselves (Bates et al., 2008). Tracking the locations of other elephants within a certain area could explain the individual variation in movement patterns and home range sizes observed for elephants (Delsink et al., 2013).

Evans, Moore & Harris (2013a) assessed if the movement patterns of reintegrated elephants differed from that of their wild counterparts. The study showed that reintegrated bulls did not display a preference for human proximity, their habitat utilisation was comparable to wild elephants, their core areas were not significantly smaller than those of wild elephants, and as time progressed, they formed part of a fission-fusion society of the wild elephants (Evans, Moore & Harris, 2013a).

Studies that have investigated the movement patterns of previously captive African elephants are sparse, especially those that have monitored released elephants over 4 years, a period deemed necessary for elephants to adapt (Lee & Moss, 1986, 2011). This gap in understanding underscores the critical need to delve deeper into these patterns to enhance the future effectiveness of elephant reintegration initiatives.

Our study was designed with the primary goal of discerning whether the flexibility and adaptability of movement behaviour of reintegrated elephants can be considered a component of determining the success of such an operation. The first aim of the study was to investigate how the elephants’ utilisation distribution areas (UDA), utilisation of the reserve, movement speeds and daily distances travelled was influenced by the rainfall season and different periods of the diurnal cycle. The second aim of our study was to investigate if the reintegrated elephants demonstrated an adaptability to seasonal changes and diurnal cycle in their ranging patterns after a period of free roaming for more than 4 years, or whether they adapted their movement patterns immediately after release. The insights drawn from this study will not only fill a significant gap in the current body of knowledge but could also guide the refinement of future elephant reintegration strategies.

Materials and Methods

The study was conducted on Shambala Private Game Reserve (SPGR) (10,000-hectare) and !Khamab Kalahari Reserve (KKR) (90,000-hectare), which are situated in the Waterberg Mountain region (Limpopo Province) and the south-eastern edge of the Kalahari (Northwest Province) in South Africa, respectively. Both reserves fall within the savanna biome, with SPGR forming part of the central bushveld bioregion and KKR forming part of the eastern Kalahari bushveld bioregion (Fig. 1) (Rutherford, Mucina & Powrie, 2006).

Figure 1 Illustration of the layout of !Khamab Kalahari Reserve and Shambala Private Game Reserves.

Map illustrating the layout of (A) !Khamab Kalahari Reserve and (B) Shambala Private Game Reserve. Map data @ 2023 Esri.

Study animals

For this study, the movement patterns of two independent, previously captive elephant groups (Table 1, Fig. 2) were observed during the different stages of reintegration onto separate fenced reserves in the Limpopo- (SPGR) and the Northwest (KKR) provinces (Table SA2).

Table 1 Summary of the elephant herd characteristics of Shambala Private Game Reserve and !Khamab Kalahari Reserve.

Characteristics	Shambala private game reserve	!Khamab kalahari reserve	
Total number of elephants on the reserve in 2024	14 (of which 10 were previously captive)	19 (of which six were previously captive)	
Founder population of the reserve:	Six of the elephants (four males and two females) were acquired from a captive elephant facility in 2002. They were captured during a culling operation which took place in the 1980’s.	The adult elephants (one adult female and four adult males (of which one died in 2016)) were captured during a culling operation in 1988.	
Place of origin:	Gonarezhou National Park in Zimbabwe.	Hwange National Park and the Zambesi Valley in Zimbabwe.	
Location of captive facility they were acquired from:	Zimbabwe.	Private reserve near Victoria Falls, Zimbabwe and moved to Letsatsing Game Park, Pilanesberg, South Africa in 2002.	
Duration in captivity:	The adult elephants on SPGR had spent over 20 years in captivity.	The adult elephants on KKR had spent approximately 28 years in captivity.	
Captive history:	The elephants were used for elephant back safaris for 12 years (2004–2016) which operated once a day for an hour in the morning. For the remainder of the day, the elephant handlers lead the elephants out into the reserve where they foraged under the guidance of the handlers and returned them back into their stables at night. The stables consisted of secure holding enclosures which were designed to separate the elephants from one another and were secured at night to prevent escape.	The elephants did not partake in any tourism activities for more than 3 h daily. There were two riding sessions conducted per day, each for 1 h. The elephants were released from their stables/paddock at 05:30 each morning, worked during the riding safaris, whereafter they were allowed to roam free and forage on the reserve under the handlers’ guidance. Following the afternoon interactive sessions, the elephants returned to their stables at 18:00–18:30.	

Figure 2 Illustration of the elephants, and the relationship between one another, that are currently on Shambala Private Game Reserve and !Khamab Kalahari Reserve.

Illustration of the elephants (origin, sex, and date of birth), and the relationship between one another, that are currently on (A) Shambala Private Game Reserve and (B) !Khamab Kalahari Reserve (SAM, Sub-adult male; YAF, Young adult female; SAF, Sub-adult female; JF, Juvenile female). (The sexes of Calf 4 and Calf 5 are unknown).

The reintegration operation

The reintegration process was uniquely tailored for the individual elephant herds of SPGR and KKR, each following its own distinctive set of stages (as detailed in Table 2). The elephants at SPGR underwent a three-stage reintegration operation. This operation spanned approximately 2 months and encompassed a Stables stage, a Boma stage, and a Release stage (refer to Table 2 for further details). Following the ceasing of commercial elephant back safaris (between 1 July 2014 and 15 July 2015), the KKR elephants remained in captivity, albeit that they were allowed to feed within the 280-hectares of Letsatsing Game Reserve (LGR) during the daylight hours under the supervision of their handlers. Their reintegration program consisted of six stages, including the translocation from LGR to KKR (Table 2).

Table 2 Summary of the different stages of reintegration of Shambala Private Game Reserve’s, and !Khamab Kalahari Reserve’s elephants into the wild.

	Stage of reintegration	Date	Description	
Shambala Private Game Reserve	1	4–16 March 2016	Stables: The elephants were not ridden; however, they were herded by the handlers during the day and locked up in individual stables, secured within an electrified fenced area (boma) at night*.	
2	17 March–20 May 2016	Boma: The elephants were moved to an electrically fenced 1.7 ha, open-air boma#. They were allowed to roam free within a 5 km radius of the boma during the day and at night, they were closed in together without being confined by individual stables.	
3	21 May	Release: The boma gate was left open and the elephants were allowed to roam free on the 10,000-hectare reserve.	
Pre-translocation stages	
!Khamab Kalahari Reserve	1 (LGR)	1 July 2014–15 July 2015	Stables/Paddock: Commercial elephant back safaris stopped, but the elephants remained in captivity. The elephants fed out on LGR under the handlers’ supervision.	
2 (LGR)	16 July–8 September 2015	Test release: During the daytime the elephants were allowed to roam free on LGR. However, they were under handler supervision who managed their movements depending on their location. Elephants were allowed to feed unattended on the reserve at night from +/− 17:00–07:00. The elephants had access to the stable/paddock area if they wanted to return, but they never utilized it.	
Translocation	9 September 2015	Translocation from LGR to KKR	
Post-translocation stages	
	3 (KKR)	10–30 September 2015	Boma: On KKR, elephant movement was managed by the handlers and was gradually reduced over time. During the day, the elephants fed on the reserve under supervision and were returned to a 2 ha, secured electrified fenced area (boma) at night (18:00–06:00)×	
	4 (KKR)	1–6 October 2015	Release (First attempt): The boma gate was opened and the elephants were allowed to roam free.	
	5 (KKR)	7 October–10 December	Due to complications with one of the bulls, the elephants were brought back to the secured electrified fenced area (boma) and continued with the boma stage to be stabilised before the second release attempt was made.	
	6 (KKR)	11 December 2015	Release (Second attempt): The boma was opened and the elephants were left to choose direction and movement themselves on the 90,000-hectare reserve. Their movements were no longer manipulated by the handlers. For the first couple of weeks lucerne (Medicago sativa) was left in the boma in case the elephants chose to return.	
Notes:

* The SPGR elephants were guided in the areas between water sources 5, 6, 7, 8 and 11 (Fig. 1E).

# An open-air boma is an electrified fenced enclosure which contains natural vegetation, shade, and water.

× The KKR elephants were guided towards four water sources (1, 4, 7 and 8) and left to forage in areas surrounding those sources (Fig. 1D).

Movement data

Prior to the reintegration of the elephants on both reserves, one of the reintegration prerequisites was the fitment of satellite-linked Global Positioning System (GPS) collars on key individual elephants to monitor their movements on an hourly basis. The key individuals (Micky-SPGR and Michael-KKR) that were collared represented the movement patterns of the two reintegrated herds on the reserves. On KKR, the study animals (Fig. 2) never separated from one another. This was confirmed by comparing initial collar data between Chikwenya and Michael, casual observations by reserve management, and through field monitoring sessions that were conducted throughout the 4-year data collection period. Following release and reintegration into the two reserves, reserve management conducted regular checks on the elephants, along with occasional sightings in between. Notably, none of the breeding herds were observed to split at any point. A similar trend was observed for the SPGR elephants, where the elephants remained together throughout the study period. The adult bull (Mickey) also never separated from the herd, which was also confirmed by previous collar data comparisons between Mouse and Mickey, and through daily observation sessions conducted by the handlers on the reserve. The fact that Michael (KKR) and Mickey (SPGR), never separated from the breeding herds is not a natural phenomenon expected for elephant bulls, However, it is likely this is reflection of the strong bonds formed during their captivity.

The collaring of key individuals was both a pre-release condition and a management decision, which presented an opportunity to study the movement patterns of these reintegrated elephants without any intervention to acquire the data to produce this study. Therefore, no ethical clearance was required for this study as data that was captured for management purposes was utilised. Iridium Satellite GPS collars manufactured by African Wildlife Tracking with a 1-h GPS fix schedule were fitted to the elephants prior to their release without the need for immobilization due to the elephants’ captive status. On the 7th of August 2020, an expired collar on the key individual at KKR was replaced by a Vertex GPS Plus collar manufactured by Vectronic Aerospace with a one-hour GPS fix. The GPS fix schedules were set in such a manner to balance between management requirements and research needs. The new GPS collar was also fitted for management purposes prior to the imminent arrival of a wild herd (September 2020) of elephants to the reserve. Immobilization was required for this fitment due to the now wild status (5 years post release) of the individual on KKR. All requirements as per the necessary regulations, Threatened or Protected Species (TOPS) legislation (Department of Environmental Affairs and Tourism (DEAT), 2007), South African Veterinary Council (Republic of South Africa, 1982), the Elephant Management Plans (EMP) (Department of Environmental Affairs and Tourism (DEAT), 2008), and the ‘National Norms and Standards for the Management of Elephants in South Africa’. Department of Environmental Affairs and Tourism (DEAT) (2008) were adhered to for this procedure.

For both elephant groups (SPGR and KKR), the data collection started when the boma gates were opened and their movements were no longer managed by the handlers (Table 2). As depicted in Table 3, the initial movement patterns of the elephants on SPGR and KKR were recorded from month 0–12 (Phase 1), consecutively. A follow-up Phase (Phase 2) was implemented to investigate how the elephants’ movement changed when they had been roaming free for more than 4 years (to align both reserves’ collar data to reflect the same period of time, we utilised the movement data transmitted 57 months post release).

Table 3 Summary of the initial movement data collected for the elephants on SPGR and KKR as well as when the elephants had been roaming free for 4 years and 8 months.

Reserve	Month 0–12 Post release movement (Phase 1)	Month 57 to 69 Post release movement (Phase 2)	
Starting date	End date	Starting date	End date	
KKR	11 Dec 2015 (Wet season)	5 Dec 2016	1 Sept 2020	31 August 2021	
GPS fix schedule	One-hourly GPS coordinate transmissions	
SPGR	21 May 2016
(Dry season)	15 May 2017	26 Feb 2021	25 February 2022	
GPS fix schedule	One-hourly GPS coordinate transmissions	

Movement behaviour of elephants

All the data acquired from the GPS collars were transmitted via satellite, downloaded through the specific collar programmes (AWT Online Tracker and GPS Plus X Collar Manager), and stored as Excel files.

The effect of seasonal changes on the elephants’ utilisation distribution areas (UDA) within each Phase of the project

To assess how the movement behaviour of the elephants changed with time and across space we calculated utilisation distributions (UD) and utilisation distribution areas (UDA). The complete movement path of each herd was subset into wet (November-February), autumn (March-April), dry (May-August), and spring (September-October) seasons (Mosase & Ahiablame, 2018; Van Der Walt & Fitchett, 2020). We then fitted dynamic Brownian bridge movement models to the seasonal paths, assuming isotopic diffusive motion between consecutive locations (Horne et al., 2007; Kranstauber et al., 2012). Dynamic Brownian bridge movement models were used to take the time dependence between successive locations, as well as the location error of the GPS points into account. Based on the temporal resolution of the locational data we specified a moving window size of 11 h and a margin of 7 h, making it possible to detect potential behavioural shifts between daytime and night-time (Kranstauber et al., 2012). Furthermore, the location error was set to 23 m for all locations (Purdon & Van Aarde, 2017). The resolution of the square raster cells was set at 30 m, the size of the moving window was set as 17 m, and the margin used for the behavioural change point analysis was set at 7 m. Finally, we calculated the 95% utilisation distribution area (UDA) (the smallest area containing 95% of the distribution) (Delsink et al., 2013) from the utilization distributions for each individual for all Phases and each season to calculate the size of the area used during the different Phases and seasons. Daily-, and hourly-displacement distances were calculated by using the continuous-time speed and distance (CTSD) estimation method (Noonan et al., 2019).

Statistical analysis

The effect of seasonal changes and diurnal cycle on the movement patterns of reintegrated elephants during Phase 1 and Phase 2 of the project

To determine if reintegrated elephants showed seasonal and diurnal variability in movement distances, we compared daily displacement distances across seasons and periods of the diurnal cycle for each herd and Phase of the project.

Four periods of the day were chosen to replicate what previous studies have chosen and comprised of dawn (05:00–07:00), midday (12:00–14:00), dusk (18:00–20:00) and night (22:00–02:00) (Kinahan, Pimm & Van Aarde, 2007; Loarie, Aarde & Pimm, 2009). For diurnal movement patterns, we calculated the mean distance travelled per minute during each period of the day. This accounted for differences in the length of periods (i.e., night encompassed 4 h whilst dawn, dusk and day were each 2 h long). The data could not be transformed to meet the assumptions of a parametric analysis and so Kruskal-Wallis’ multiple comparison with Dunn’s post-hoc test was used to identify differences in travel distances throughout the diurnal cycle.

Total daily travel distances were calculated for each season within each of the two Phases. Data were log-transformed to meet the assumption of normality required for parametric testing. We used a one-way ANOVA with Tukey’s post-hoc test to identify seasonal differences in travel distances for each herd, within each Phase of the project.

Comparison between the elephants’ movement patterns of the elephants between Phase 1 and Phase 2 of the project

To understand whether reintegrated elephants showed the ability to adapt their movement patterns over time, we compared their seasonal and diurnal movement cycles immediately after (Month 0–12), and 57 months post-reintegration.

For mean distances travelled within each period of the diurnal cycle, data could not be transformed to meet the assumptions of parametric analyses, so a Friedman test was used instead to test for differences between Phases only. For seasonal total travel distances, data were log-transformed to meet the assumptions of normality required for a two-way ANOVA, considering Phase, season, and the interaction between the two effects.

All analyses were implemented in the R statistical computing environment (R Core Team, 2022) along with the package ‘move’ (Kranstauber, Safi & Bartumeus, 2014). All utilisation distribution maps were created using ArcGIS Pro 3.1.0 and visually analysed and interpreted.

Results

During the study, a total of 16,596 GPS locations were recorded for the SPGR elephants (8,205 during Phase 1; 8,391 during Phase 2), and a total of 15,911 GPS locations were recorded for the KKR elephants (7,315 during Phase 1; 8,596 during Phase 2). No collar failures were observed during the data collection session.

The effect of seasonal changes on the elephants’ utilisation distribution (UD) and areas (UDA) within Phase 1 and Phase 2 of the project

Visual inspection of the utilisation distribution (UD) and area (UDA)

The elephants on SPGR had the largest UDA during the wet season, followed by autumn during both Phases of the project (Fig. 3). The elephants had smaller UDA in the dry season during both Phases (Fig. 3). During Phase 1, their UDAs varied between 3,172.78-hectare to 5,964.77-hectare. The elephants exhibited the largest UDA during the wet season (5,964.77-hectare) where they utilised 60% of the reserve (Fig. 3). During Phase 2, the elephants also had the largest UDA during wet season (6,054-hectare), followed by autumn (5,357-hectare) and spring (4,994-hectare) and the smallest UDA during the dry season (4,132-hectare) (Fig. 3).

Figure 3 The utilization distributions that were observed during the dry, spring, wet and autumn season, 12 months post release, and when the elephants on SPGR have been roaming free for 57 months.

Illustration of the utilization distribution area (UD—hectares) and utilisation distributions (high and low use) during the dry, spring, wet and autumn season, (A) 12 months post release (Phase 1) and (B) when the elephants on SPGR had been roaming free for 57 months (Phase 2). Map data @ 2023 Esri.

Season influenced the KKR elephants’ UDAs during Phases 1 and 2 (Fig. 4). The UDA of the elephants was larger during the wet season for both Phases 1 (7,310.08-hectare) and 2 (5,460.61-hectare) when compared to the other seasons (Fig. 4). During Phase 1, the elephants had the largest UDA (8,270.56-hectare) during the dry season and the smallest during autumn (2,068.13-hectare). A contradictory trend was observed during Phase 2, where the elephants’ UDA was the smallest during the dry season (3,587.20-hectare) (Fig. 4).

Figure 4 The utilization distributions that were observed during wet, autumn, dry, and spring seasons, 12 months post-release and when the elephants have been free roaming on KKR for 57 months.

Illustration of the Utilization distributions area (UD—hectares) and the utilisation distributions (high and low use) observed during wet, autumn, dry, and spring seasons, (A) 12 months post release and (B) when the elephants have been free roaming on KKR for 57 months. Map data @ 2023 Esri.

When we investigated the UDA across the reserves during Phase 1 and 2 (Figs. 3 and 4), it was noted that both the SPGR and KKR elephants changed their patterns depending on the season they were in.

The effect of diurnal cycle and season on the movement speeds and daily distance travelled by the reintegrated elephant during Part 1 and 2 of the project

Diurnal cycle and movement speeds

It is evident that time of day had a significant effect on the speed travelled by the SPGR elephants during Phases 1 (Kruskal-Wallis Chi-squared = 309.99, df = 3, p-value < 0.0001) and Phase 2 (Kruskal-Wallis Chi-squared = 379.79, df = 3, p-value < 0.0001) of the project. During both Phases, the elephants travelled at significantly (p < 0.0001) faster speeds during the day (12:00–14:00) and at dusk (18:00–20:00) than at dawn (05:00–07:00) and at night (22:00–02:00). The elephants also travelled at significantly faster speeds during dawn than at night (p < 0.0001) (Fig. 5; Table 4).

Figure 5 SPGR elephants’ average moving speed (m/min) 12 months days post release (Phase 1) compared to when free roaming for 57 months on SPGR (Phase 2).

SPGR elephants’ average moving speed (m/min) (A) 12 months post-release (Phase 1) compared to when free roaming for (B) 57 months on SPGR (Phase 2). The crosses represent the means, whereas the central horizontal bars are the medians. The first and third quartiles are the lower and upper limits of the box, respectively. The whiskers represent the minimum and maximum values. The points above or below the whiskers’ upper and lower bounds may be considered outliers (the blue dots represent outlier group 1 (>1.5 interquartile ranges from the median), blue stars represent outlier group 2 (>3 interquartile ranges from the median), and the grey diamonds represent the minimum and maximum data points). Different letters within each Phase represent significant differences between the average movement speeds exhibited across the four time periods (N = number of observation days).

Table 4 Summary of the mean (+SD) movement speeds (m/min) travelled by the SPGR and KKR elephants during the different time periods of the day during Phase 1 and 2 of the project (N = Number of observations).

Reserve	Phase	Dawn	Midday	Dusk	Night	
SPGR	Phase 1	5.041a	5.761b	5.204b	3.355c	
(SD = 5,539)	(SD = 5,380)	(SD = 4.738)	(SD = 4.740)	
(N = 695)	(N = 701)	(N = 706)	(N = 1174)	
Phase 2	4.866a	5.942b	6.065b	3.215c	
(SD = 4.080)	(SD = 5.251)	(SD = 4.920)	(SD = 3.990)	
(N = 730)	(N = 730)	(N = 730)	(N = 1092)	
KKR	Phase 1	7.104a	4.321b	9.594c	8.466d	
(SD = 9.707)	(SD = 5.418)	(SD = 10.413)	(SD = 9.686)	
(N = 597)	(N = 560)	(N = 640)	(N = 1237)	
Phase 2	4.510a	3.133b	8.859c	8.373d	
(SD = 6.549)	(SD = 3.557)	(SD = 6.572)	(SD = 9.531)	
(N = 721)	(N = 709)	(N = 716)	(N = 1426)	
Note:

Different letters within each Reserve and Phase represent significant differences between the average movement speeds exhibited across the four time periods

For the SPGR elephants, one peak in average distance travelled per hour was observed between 06:00–08:00 during Phase 1, followed by a decrease at midday (12:00–14:00), and another increase at 23:00. During Phase 2, two peaks were observed between 07:00–11:00, 17:00–18:00 and another increase at 23:00 (Fig. 6).

Figure 6 SPGR elephants’ average hourly displacements 12 months post-release compared to when they have been free roaming on SPGR for 57 months.

SPGR elephants’ average hourly displacements (+SD) 12 months post release (Phase 1) compared to when free roaming on SPGR for 57 months (Phase 2).

The speeds travelled by the KKR elephant across all four time periods differed significantly within Phase 1 (Kruskal-Wallis Chi-squared = 195.97, df = 3, p-value < 0.0001) and Phase 2 (Kruskal-Wallis Chi-squared = 599.93, df = 3, p-value < 0.0001) of the reintegration (Fig. 7; Table 4). During both Phases of the project, the elephants travelled at faster speeds during dusk (18:00–20:00), followed by nighttime (22:00–02:00). The elephants’ movements were significantly faster during dusk than night (p < 0.0001), dawn (p < 0.0001), and day (p < 0.0001) during Phases 1 and 2 of the reintegration (Fig. 7).

Figure 7 KKR elephants’ average moving speed (m/mins) 12 months post-release (Phase 1) compared to when they had been free roaming for 57 months on KKR (Phase 2).

KKR elephants’ average moving speed (A) 12 months post release (Phase 1) compared to when (B) free roaming for 57 months on KKR (Phase 2). The crosses represent the means, whereas the central horizontal bars are the medians. The first and third quartiles are the lower and upper limits of the box, respectively. The whiskers represent the minimum and maximum values. The points above or below the whiskers’ upper and lower bounds may be considered as outliers (the blue dots represent outlier group 1 (>1.5 interquartile ranges from the median), blue stars represent outlier group 2 (>3 interquartile ranges from the median), and the grey diamonds represent the minimum and maximum data points). Different letters within each Phase represent significant differences between the average movement speeds exhibited across the four time periods (N = number of observation days).

The KKR elephants travelled larger distances at dusk (18:00–20:00), dawn (05:00–07:00), and night (22:00–02:00) in comparison to the distances travelled at midday (12:00–14:00) during Phase 1 (Fig. 8). Only one peak occurred during Phase 1 between (18:00–22:00). During Phase 2, there was an increase in movement at dusk (18:00–20:00), followed by a decrease at night (22:00–02:00). At dawn (05:00–07:00), there was another slight increase in movement and the lowest distances were travelled at midday (12:00–14:00) (Fig. 8).

Figure 8 KKR elephants’ average hourly displacements travelled 12 months post-release compared to when free roaming for 57 months on KKR.

KKR elephants’ average hourly displacements (+SD) travelled 12 months post release (Phase 1) compared to when free roaming for 57 months on KKR (Phase 2).

Daily distance travelled

When the influence of season and Phase on the average daily distance travelled by the SPGR (F = 10.240, df = 3, p < 0.0001) and KKR (F = 27.704, df = 3, p < 0.0001) herd was investigated, a significant interaction was noted. Thus, the elephants’ seasonal movement patterns differed depending on the Phase they were in.

The SPGR elephants’ average daily displacement peaked during the wet season and was the lowest during the dry season in Phase 1 (Fig. 9). The average daily distances travelled during the wet season was significantly larger than those observed during autumn (p = 0.003), spring (p = 0.0055), and dry season (p = 0.0001) during Phase 1. During Phase 2, they travelled significantly larger distances during spring and autumn than during wet, (spring: p = 0.006; autumn: p = 0.043) and dry season (spring: p = 0.000; autumn: p = 0.004) (Fig. 9).

Figure 9 The daily displacement exhibited by the SPGR elephants during the dry, autumn, spring, and wet season 12 months post-release compared to when free roaming for 57 months.

The daily displacement exhibited by the SPGR elephants during the dry, autumn, spring, and wet season 12 months post release (Phase 1) compared to when free roaming for 57 months (Phase 2). The red crosses represent the means, whereas the central horizontal bars are the medians. The first and third quartiles are the lower and upper limits of the box, respectively. The whiskers represent the minimum and maximum values. The points above or below the whiskers’ upper and lower bounds may be considered as outliers (the hollow circles represent outlier group 1 (>1.5 interquartile ranges from the median), the stars represent outlier group 2 (>3 interquartile ranges from the median), and the diamonds represent the minimum and maximum data points). Different letters within each Phase represent significant differences between the daily displacement exhibited (m/day) across the four seasons (N = number of observations days).

During Phase 1, the KKR elephants travelled the largest daily distances during the dry season, followed by the wet season. The distances travelled during the wet and dry season differed significantly (p = 0.034) during Phase 1. Furthermore, the distances travelled during the dry season were also significantly larger than that recorded during autumn (p < 0.0001) and spring (p = 0.026) (Fig. 10; Table 5). A contradicting trend was observed during Phase 2 of the reintegration where the elephants travelled significantly larger distances during autumn and spring than during the dry (autumn: p = 0.022; spring: p = 0.002) and wet (autumn: p = 0.049; spring: p = 0.006) season (Fig. 10; Table 5).

Figure 10 The daily displacement of the KKR elephants during dry, autumn, spring, and wet season 12 months post-release compared to when free roaming for 57 months.

The daily displacement of the KKR elephants during dry, autumn, spring, and wet season 12 months post release (Phase 1) compared to when free roaming for 57 months (Phase 2). The red crosses represent the means, whereas the central horizontal bars are the medians. The first and third quartiles are the lower and upper limits of the box, respectively. The whiskers represent the minimum and maximum values. The points above or below the whiskers’ upper and lower bounds may be considered as outliers (the hollow circles represent outlier group 1 (>1.5 interquartile ranges from the median), the stars represent outlier group 2 (>3 interquartile ranges from the median), and the diamonds represent the minimum and maximum data points). Different letters within each Phase represent significant differences between the daily displacement exhibited (m/day) across the four seasons (N = number of observations).

Table 5 Summary of the mean (+SD) daily distance travelled by the SPGR and KKR elephants during the different seasons during Phase 1 and 2 (N = number of observation days).

Reserve	Phase	Dry	Autumn	Spring	Wet	
SPGR	Phase 1	6,355.46a	6,389.09a	6,415.13a	7,578.52b	
(SD = 2,072.45)	(SD = 2,306.09)	(SD = 1,953.98)	(SD = 2,636.70)	
(N = 118)	(N = 61)	(N = 61)	(N = 120)	
Phase 2	6,479.98a	7,563.94b	7,666.01b	6,836.26a	
(SD = 1,465.74)	(SD = 2,190.76)	(SD = 1,731.01)	(SD = 2,024.93)	
(N = 123)	(N = 61)	(N = 61)	(N = 120)	
KKR	Phase 1	11,518.88a	6,370.64b	9,481.39c	10,445.20c	
(SD = 4,375.71)	(SD = 3,361.02)	(SD = 3,789.95)	(SD = 5,446.65)	
(N =123)	(N = 61)	(N = 61)	(N = 115)	
Phase 2	7,960.94a	9,661.34b	10,081.10b	8,360.92a	
(SD = 2,640.32)	(SD = 4,456.54)	(SD = 3,840.56)	(SD = 3,484.37)	
(N = 123)	(N = 61)	(N = 61)	(N = 120)	
Note:

Different letters within each Phase represent significant differences between the average movement speeds exhibited across the four time periods.

Comparison between the elephants’ utilisation distributions (UD) and areas (UDA), as well as the movement patterns between Phase 1 and Phase 2 of the project

Utilisation distributions (UD) and areas (UDA)

As mentioned in the previous section the SPGR elephants exhibited similar UDAs during both Phases of the project across all seasons, however, during the spring season, the elephants exhibited larger UDAs during Phase 2 of the project (Fig. 3). During Phase 1, the KKR elephants exhibited larger UDAs during the wet and dry seasons, and smaller UDAs during autumn and spring season than that noted during Phase 2 (Fig. 4).

When the utilisation distribution of the SPGR elephants were compared between Phases 1 and 2, no noticeable trend was observed (Fig. 3). During the dry season, the SPGR elephants utilised more sections close to the southern fence line during Phase 2 than Phase 1. A contradictory trend was observed during the wet season, where the SPGR elephants utilised sections bordering the western fence line during Phase 2, whereas they utilised sections bordering the eastern fence line during Phase 1 of the project. We observed similar utilisation distributions during Phases 1 and 2 during spring (Fig. 3).

The KKR elephants utilized similar areas during the four seasons when movements during Phases 1 and 2 of the project were compared (Fig. 4). During Phase 1 (Fig. 4A), the elephants explored a small section in the northern part (areas close to water source 12) of the reserve when they were just released (2015, wet season), as well as during their first dry season (2016, dry season). However, they did not utilize this section when they had been roaming free for more than 57 months (Fig. 4B). During spring season (2020), the elephants investigated areas bordering the western fence line northwest of water source 2. They also travelled further east, towards water source 5, which they did not utilize during Phase 1 of the project. The elephants began to explore areas close to water source 6 during the dry season (2021). During Phase 2, the elephants travelled through areas bordering one of the eastern fence lines during wet, autumn, dry and spring seasons.

Diurnal cycle and movement speeds

The movement speed (m/min) of SPGR elephants did not differ significantly between Phases 1 (Fig. 5A) and 2 (Fig. 5B) of reintegration when the period of the diurnal cycle was considered (Friedman chi-squared = 0, df = 1, p-value = 1). The movement speed (m/min) of KKR elephants differed significantly between Phases 1 (Fig. 7A) and 2 (Fig. 7B) of reintegration when accounting for time of day (Friedman chi-squared = 4, df = 1, p-value = 0.0455). Elephants on average moved faster in Phase 1 compared to Phase 2.

Daily distance travelled

The SPGR elephants travelled significantly (F = 4.654, df = 1, p < 0.001) larger daily distances during Phase 2 of the reintegration than during Phase 1. The KKR elephants travelled significantly (F = 8.882, df = 1, p < 0.001) larger daily distances during Phase 1 of the reintegration than during Phase 2.

Discussion

The study provided a unique opportunity to showcase the adaptability and flexibility of two previously captive elephant herds and how their movement evolved several years after their reintegration into the wild. As mentioned by various researchers (Lee & Moss, 1986, 2011; Druce, Pretorius & Slotow, 2008), elephants require several years to adapt to new environments and individuals’ initial movement patterns will differ, especially after a translocation event (Tiller et al., 2022) as was the case with the KKR elephants.

The elephant herds on SPGR and KKR both shifted their utilisation distributions across the reserves, as well as the areas (UDA) thereof, as the seasons changed. Both the SPGR and KKR elephants exhibited larger UDAs during the wet season during both Phases, which is similar to that reported for wild elephants. This was a positive observation, as some researchers have questioned the ability of captive elephants to adapt to change such as needing to adapt back to their natural environments (Thulin & Röcklinsberg, 2020). This change in seasonal ranging patterns was expected as previous studies have shown that elephant movement is often dictated by rainfall season, temperature, and resource distribution (Kinahan, Pimm & Van Aarde, 2007). During the wet season, elephants tend to range further and consequently exhibit larger home range sizes (Leggett, 2006; Young, Ferreira & Van Aarde, 2009) due to their decreased dependency on water during this season (Smit, Grant & Devereux, 2007; Wittemyer et al., 2007; Young, Ferreira & Van Aarde, 2009; Chamaillé-Jammes et al., 2013). Even though the season had a significant effect on the SPGR and KKR elephants’ ranging patterns, it is important to note that both reserves have artificial water sources. Consequently, access to these artificial water holes can increase dry season ranging patterns of elephants that are not restricted by fences. This allows the elephants to range further towards areas that would have only been accessible during the wet season (Viljoen, 1989; Smit, Grant & Devereux, 2007; de Beer & van Aarde, 2008; Thomas, Holland & Minot, 2008; Purdon & Van Aarde, 2017). The presence of these artificial water sources can decrease the differences observed between wet and dry season movement patterns (Loarie, Aarde & Pimm, 2009). Additionally, the elephants’ unfamiliarity with their new environment during Phase 1 could have overshadowed any of these environmental factors.

Utilisation distribution area (UDA)

The UDAs of wild elephants differ depending on the habitat type (Douglas-Hamilton, 1972; Douglas-Hamilton, Krink & Vollrath, 2005; Dolmia et al., 2007), rainfall season (Leggett, 2006; Garstang et al., 2014; MacFadyen et al., 2019), individual elephants (Delsink et al., 2013), and their familiarity with a certain area (Druce, Pretorius & Slotow, 2008; Tiller et al., 2022). In fenced areas, their home ranges are often small (1,000–8,000-hectare) (Douglas-Hamilton, Krink & Vollrath, 2005; Leggett, 2006; Dolmia et al., 2007; Ngene et al., 2017). The home range sizes in relation to the total reserve size also varies between different elephant herds. It has been reported that breeding herds on Pongola Game Reserve exhibited a maximum area of range of 3,670-hectare (reserve size = 7,360-hectare) which is 50% of the available space (Shannon et al., 2006). Bulls on Pilanesberg National Park tend to have a range of 9,970-hectare (reserve size = 50,000-hectare) which results in 20% of the reserve utilized (Slotow & van Dyk, 2004). Addo National Park is considered a large-fenced system and researchers have shown mean range size for females is 5,500-hectare (reserve size = 70,000-hectare), which amounts to 8% of the reserve being utilized (Whitehouse & Schoeman, 2003). The UDAs noted for the SPGR elephants ranged between 3,173–5,964-hectare during Phase 1 (32–60% reserve utilisation) and 4,132–6,054-hectare during Phase 2 of the reintegration (41–60% reserve utilisation). The UDAs reported for the KKR elephants ranged between 2,068–8,271-hectare during Phase 1 (2–9% reserve utilisation) and 3,587–7,310-hectare during Phase 2 of the project (4–8% reserve utilisation). The UDAs and percentage of the reserves utilised reported for both the reintegrated elephant herds on the two reserves were similar to that observed for wild elephants in small- and large-fenced areas (Whitehouse & Schoeman, 2003; Slotow & van Dyk, 2004; Shannon et al., 2006)

Utilisation distribution

When the seasonal utilisation distribution of the SPGR elephants were investigated, it was evident that the elephants started off by exploring small areas of the reserve during the first two seasons (dry and spring) post-release, which is considered normal exploratory behaviour of wild elephants (Druce, Pretorius & Slotow, 2008). When the SPGR elephants were still in captivity, they roamed in the areas between water sources 5, 6, 7, 8 and 11 under the guidance of their handlers (Fig. 1E). The elephants did not remain in that section of the reserve for a long period of time and began to explore unfamiliar sections of the reserve, which could be confirmed as a positive observation.

The KKR elephants explored a large section in the centre of the reserve during their first season (wet season) post-release (Fig. 4). Prior to their release (during the boma stage), the elephants were guided towards four water sources (1, 4, 7 and 8) (B. Mitchell, 2015, personal communication) and left to forage in areas surrounding those sources (Fig. 1D). Even though they were presented with the opportunity to utilize water sources 7 and 8, they never visited those sources again after release and when their movements were no longer influenced by their handlers. This could indicate that the elephants could distinguish between various food and water sources and established preferences based on the knowledge that they had gained. The elephants also travelled further distances daily during their first months as free elephants (Phase 1). This emphasises the importance of choosing the optimal release date, taking into account the season and other environmental factors. For example, releasing the elephants during the wet season allows for fewer constraints on movement and will enable the elephants to explore more freely. During their first dry season post-release (Phase 1), the KKR elephants’ UDAs and average daily distance travelled were larger than that noted for all the other seasons. During the dry season, the density of water sources were less, forcing them to travel longer distances to locate new water sources and settle once they have found adequate sources. However, if they did not locate such sources, they either had to return to sources they knew or increase their distance of movement to search for other sources. The KKR elephants utilized similar areas during Phase 2 when compared to Phase 1, across the four seasons (Fig. 4). However, during Phase 2 the manner in which they utilized the reserve was more effective, as they spent more time within certain areas. Once the elephants were free roaming (Phase 2), they began exploring new sections of the reserve and utilised seasonal water sources. This could indicate an increase in knowledge of how seasonal rainfall affects the water sources and surrounding foraging areas. The translocation of a wild herd in September 2020, could also have contributed towards the expansion into these novel areas, as they have been observed together. Due to elephants’ ability to remember the spatial location of other elephants (Bates et al., 2008), they possibly wanted to either avoid competition (Dunbar, 1992) or move together as one herd due to increased security (Hamilton, 1971), demonstrating social learning (Lee & Moss, 1999) and fusion patterns (Goldenberg et al., 2022).

Diurnal cycle and movement speeds

It has also been reported that elephants shift their activity peaks towards cooler times of the day to avoid thermal stress (Kinahan, Pimm & Van Aarde, 2007; Kinahan et al., 2007). The more prominent peaks in movement of the SPGR elephants during Phase 2, could indicate that after a couple of years, the elephants have gained more knowledge of their environment and have adjusted their behavioural strategies to meet their physiological needs (thermoregulation), which they could not achieve in captivity due to the handlers managing their movement thus restricting the ability to thermoregulate correctly. The results also showed that the speed at which the SPGR elephants travelled during both Phases 1 and 2 of the reintegration, was significantly faster during mid-day and dusk than at dawn, and the slowest during night-time (Fig. 5). Literature has shown that elephants often visit water sources between 18:00–20:00 during dry conditions (Valeix, Chamaillé-Jammes & Fritz, 2007). Therefore, elephants walk at relatively high speeds (Chamaillé-Jammes et al., 2013) under the afternoon sun, to reach water at the end of the day. The lower movement speed during dawn and night could indicate that the elephants forage during those time periods as research has confirmed that lower displacement speeds can reflect foraging or resting behaviour (Morales et al., 2004; Purdon, 2015).

During both Phases, the KKR elephants showed a slight increase in their movement at dawn (05:00–07:00), dusk (18:00–20:00) and had the lowest movement at mid-day (12:00–14:00), which has also been reported for wild African elephants (Loarie, Aarde & Pimm, 2009; Mole et al., 2016). This herd preferred to cross the open pans to water during dusk and night-time, which possibly is attributed to feeling more secure under the cover of darkness. During Phase 1, their movement remained high between 22:00–05:00. This larger hourly displacement and faster movement speeds observed during night-time during Phase 1 could be attributed to the vegetation type, temperature, and the herd characteristics. Studies have shown that increased diurnal anthropogenic or other species disturbance could result in animals shifting their movement patterns to become more nocturnal (Lima & Dill, 1990; Wrege et al., 2010; Ihwagi et al., 2018). These factors could have caused the elephants to feel more vulnerable during the day and reluctant to travel large distances without the cover of darkness, especially in the presence of predators (Lima & Bednekoff, 1999; Bleicher, 2017). However, during Phase 2, the elephants’ movement decreased after 00:00, which is considered to be more natural and shows that elephants require a certain amount of time to adapt to a new environment (Loarie, Aarde & Pimm, 2009; Goldenberg et al., 2022).

Daily distance travelled

The SPGR and KKR elephants travelled the furthest distances daily during autumn and spring. Both these seasons are transitions from wet to dry and vice versa. The reserves’ habitat comprises a vast variety of food resources (different shrub-, tree- and grass species). These food resources will inherently be at optimum grazing physiological stage during different times of the year and will therefore result in animals having to travel different distances throughout the seasons (Johnson, Parker & Heard, 2002; Provenza et al., 2003). During spring, new leaves sprout on the trees, which could result in further distances travelled to gain access to this resource. During autumn, the trees lose their leaves and force the elephants to move further distances to utilise other sources to meet their nutritional demands. Temperature is another important driver of elephants’ daily movement patterns as it determines which strategies are used to thermoregulate during warm and cold periods of the day (Purdon, 2015; Wato et al., 2018).

Conclusions

The primary aim of the study was to discern whether flexibility and adaptability of movement patterns of reintegrated elephants could be utilised to assess the success of that particular operation. The study showed that the reintegrated elephants were able to adjust their movement patterns in response to diurnal and seasonal variability, and over time (i.e., immediately after versus 4-years post reintegration). We also aimed to evaluate their adaptability by utilizin gexisting literature on their wild counterparts as a foundation for comparison. The results showed differences between seasonal movement patterns (daily displacement), seasonal utilisation distribution areas (UDA), as well as areas utilised across the reserves for two reintegrated elephant herds. As hypothesised, the UDAs of both herds were comparable to that reported by various other researchers and both herds exhibited large home range sizes during the wet season. Additionally, the KKR and SPGR elephants adapted their speed of movement and distances travelled during different time periods of the day. It is evident that the elephant herds on both reserves used different adaptation strategies, which was expected as they were exposed to unique reintegration programs and unique conditions. A factor that might have contributed towards their different adaption strategies was their knowledge of the reserve prior to the reintegration and the access to wild elephants. The SPGR elephants had knowledge of certain areas of the reserves prior to their release, whereas the KKR elephants were translocated onto a new reserve prior to their release and only had access to a wild herd during Phase 2 of the study. Regardless, our study has shown that both herds have adjusted their movement patterns to environmental or ecological factors on the reserve.

Lastly, both elephant herds, having been in captivity for 15–28 years, were able to acquire knowledge and alter their movement strategies over the years roaming as free elephants. Reflecting on what could have contributed towards the resilience of these reintegrated elephants, we found that appropriate release date (season), available habitats and space on the reserves played a substantial role. Additionally, the reintegration process implemented for each herd was unique and ensured that the elephants had adequate skills to explore without the presence of human interaction. Lastly, the reintegration of entire social groups has proven to increase behavioural flexibility and adaptability of the herds.

Supplemental Information

Supplemental Information 1 Summary of the elephants that have been reintegrated onto fenced reserves in South Africa.

Supplemental Information 2 Summary of the reserve characteristics of Shambala Private Game Reserve and !Khamab Kalahari Reserve.

Supplemental Information 3 Data analysis of the movement data of two reintegrated African elephant herds.

We compared daily displacement distances across the four seasons and periods of the diurnal cycle (Day, Dusk, Dawn, and NIght) for each herd (KKR and SPGR) and phase (1 and 2). We calculated the total travel distances for each season and log-transformed the data to meet the assumption of normality required for parametric testing. We used a one-way ANOVA with Tukey’s post-hoc test to identify seasonal differences in travel distances for each herd and phase. For diurnal movement patterns, we calculated the mean distance travelled per second for each period of the day. These data could not be transformed to meet the assumptions of a parametric analysis and so Kruskal-Wallis’ multiple comparison with Dunn’s post-hoc test was used to identify differences in travel distances throughout the diurnal cycle.

Supplemental Information 4 Calculating movement Metrics of two reintegrated african elephant herds.

The speed of movement, distance traveled, and angles moved were calculated for each data point that was linked to a unique time and date stamp.

Supplemental Information 5 Dynamic Brownian Bridge Movement Model.

Dynamic Brownian bridge movement model (dBBMM) was used to calculate the home range size of each herd within the four separate seasons (wet, dry, autumn and spring).

This method takes into account the time dependence between successive locations and the location error (set at 23m) of the GPS points.

Supplemental Information 6 Creating and caluculating home ranges with the most simple Minimum Convex Polygons (MCP) method and using the kernel estimation method to produce utilization distributions of the two elephant herds.

We started off with calculating the most simple MCP and ended off with calculating HR using a dynamic Brownian bridge movement model. Under the kernel estimation and utilization distribution (UD) model, we consider that the animal’s use of space can be described by a bivariate probability density function, the UD, which gives the probability density to relocate the animal at any place according to the coordinates (x, y) of this place.

Supplemental Information 7 The GPS coordinates were obtained from the satellite-linked Global Positioning System (GPS) collars fitted onto two sperate elephant herds over the study period.

Each data point represents a GPS location (longitude and latitude) of the two elephant herds on the two separate reserves. Each data point is also linked to an exact time and date stamp. The date stamps were used to categorize the data into 4 seasons (Dry, Wet, Autumn, and Spring) (Tab 1). The home range sizes (Tab 2) and total daily distances traveled by the elephants were also calculated (Tab 4 and 5).

We would like to thank TJ Steyn, owner of Shambala Private Nature Reserve (SPGR), as well as !Khamab Kalahari Reserve (KKR) for granting us access to the collar data of their elephant herds. Thank you, Brett Mitchell for the creation and implementation of the reintegration programs, as well as the elephant monitors at SPGR and KKR who assisted during the reintegration programs. Lastly, we would like to thank the anonymous reviewers for their valuable contribution towards the final edition of this manuscript.

Additional Information and Declarations

Competing Interests

Author Contributions

Data Availability

Audrey Delsink is the Wildlife Director of the Humane Society International-Africa. Hanno Kilian is the Ecologist at !Khamab Kalahari Reserve. Andrew Purdon is employed by M.A.P Scientific Services.

Tenisha Roos conceived and designed the experiments, performed the experiments, prepared figures and/or tables, authored or reviewed drafts of the article, and approved the final draft.

Andrew Purdon conceived and designed the experiments, analyzed the data, prepared figures and/or tables, authored or reviewed drafts of the article, and approved the final draft.

Victoria Boult conceived and designed the experiments, analyzed the data, authored or reviewed drafts of the article, and approved the final draft.

Audrey Delsink performed the experiments, authored or reviewed drafts of the article, and approved the final draft.

Brett Mitchell conceived and designed the experiments, performed the experiments, authored or reviewed drafts of the article, and approved the final draft.

Petrus Johannes Kilian performed the experiments, authored or reviewed drafts of the article, and approved the final draft.

The following information was supplied regarding data availability:

The raw data and code are available in the Supplemental Files.

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
