# Peer review of "Movement patterns of two reintegrated African elephant (Loxodonta africana) herds: transitioning from captivity to free-living"

_PeerJ, doi:10.7717/peerj.17535_

## Round 0.1 · original submission · Major Revisions

Two of three reviewers have major reservations about your manuscript, and the third has minor comments. Given this, I would like to see a major revision dealing with all of the comments. Please revise paying particular attention to the more critical comments.

Once again, thank you for submitting your manuscript to PeerJ and we look forward to receiving your revisions.

·

Basic reporting

The English is clear and professional . I made a few grammatical comments here and there in the text or in the review.
The structure, figures, data curation are professional and relevant to the questions. The references are appropriate

Experimental design

This is an original primary research and is an important addition to the very scarce studies on reintegrating captive elephants to the wild.
Methods are clear and can be replicated with the raw data.

Validity of the findings

The findings are important for the future of reintegrating elephants back to the wild,showing clearly that this is possible. It is the first study on the adaptation to freedom for two separate elephant herds. The conclusions are appropriate and to the point.

Additional comments

Reviewer comments to manuscript
Movement Patterns of Two Reintegrated African Elephant (Loxodonta africana) Herds: Transitioning from Captivity to Free-Living
Tenisha Roos, Andrew Purdon, Victoria L. Boult, Audrey Delsink, Brett Mitchell, Hanno Kilian

This is an important further study on reintegration of former captive elephants, to add to the rather sparce literature to date. The structure and experimental design, and the validity of findings are appropriate and well structured. The research questions are well defined in the introduction and the conclusions are linked and limited to the research question and results. The references are appropriate.

General Comments:
Here and there the grammar needs to be revised, but I made comments each time.
Sometimes you write a) and sometimes (a), which one is correct? I have not marked each time, so please check what the Journal requirements are and be consistent.


Abstract
Line 20 Given the escalating challenges in captive elephant management,
I would prefer the first word to be “ With the escalating challenges….
Just sounds better?

Line 32 ……reintegrated elephants demonstrated an adaptability
Adaptability to what, the seasons? Environment? water points?

Line 54 Although not new in southern Africa,
What is not new, the studies or the reintegration? Unclear
Perhaps something like: “Although reintegration is not new in SA….”

Line 58 Captive facilities are often ill-equipped to manage unwanted behaviours such as aggression and frustration, and to mitigate risk, may stop handling animals …
Something grammatically not quite right here. The captive facilities ….. stop handling…?
“Keepers may stop handling….” Or in passive mode: “They are not handled any more or restrained further.” or similar construction


Line 62 …, leading to the conclusion that an animal is unmanageable
Not sure what you want to say here. If it makes the animal unmanageable, then say so. “Leading to the conclusion” could also imply wrong conclusion, or a suggestion

Line 71
You may want to add one or two references e.g.:
Chemical: Rasmussen LEL & Schulte B. 1998. Chemical signals in the reproduction of Asian and African elephants. Anim. Reprod. Sc. 53: 19-34
Auditory: McComb K, Moss C, Sayialel S, Baker L. 2000. Unusually extensive networks of vocal recognition in African elephants. Anim Behav. 59;1103-1109

Line 74 (Mumby & Plotnik, 2018)
Repetition??

Line 80 -82
You could add Goldenberg et al rather than Asian elephants:
I think you may find this paper relevant to your manuscript, and I feel you could integrate some of the results as comparison to yours, or at least reference it. It’s one of the few on African elephants available.
Goldenberg SZ, Hahn N, Stacy-Dawes J, Chege SM, Daballen D, Douglas-Hamilton I,
Lendira RR, Lengees MJ, Loidialo LS, Omengo F, Pope F, Thouless C, Wittemyer G and Owen MA (2021) Movement of Rehabilitated African Elephant Calves Following Soft Release Into a Wildlife Sanctuary. Front. Conserv. Sci. 2:720202.
doi: 10.3389/fcosc.2021.720202

Is your behaviour paper not in print or in review?? Then you could add it?

Line 90 The success of reintegration operations hinges on several factors, including the social dynamics of the elephant group, the age of the elephants, the method of reintegration, and the characteristics of the release site.
What about the history of captivity?

Line 93 Goldenberg et al 2021????

Line 97 The consensus is that adolescence is the optimal life stage
As it does not seem the best for elephants, maybe don’t use the word consensus, but rather say in some species as your references all refer to other species. I doubt that reintegrating a herd of adolescent elephants will be optimal.

Line 105 Unbalanced bull hierarchies resulting in unusual elephant musth patterns, have been proven to contribute towards behavioural abnormality such as fatal encounters between elephants and white rhinoceros
This implies there were fatalities on both sides, but it was not the case, only rhinos were killed, so perhaps rephrase.

Line 150 This ability could explain the individual variation in movement patterns and home range sizes observed for elephants
Does “this” refer to everything or only the last mentioned remembering location of herd members?? Not clear
Maybe: These abilities???

Line 171 …. we wanted to determine whether they aligned with what literature has shown for their wild counterparts
I find this phrasing a bit weird, maybe rephrase something like: whether they aligned with the patterns shown in literature for their wild counterparts ?

Line 172 Wild elephants show seasonal and diurnal variation in their movement patterns such as larger home ranges and greater displacement distances in the wet season.

This is a repetition from above, not sure if it should stay or not. Might depend on the editor

Line 176 and how the utilisation of the reserve was exploited
Exploited sounds negative and utilisation of the entire reserve is positive, please find a better wording


212 Table 1
Captive history SPGR The elephants conducted elephant back safaris for 12 years

I think its rather people than the elephants that conduct something, perhaps say the elephants were used for EBS, or a similar formulation?

For the remainder of the day, the elephant handlers lead the elephants out into the reserve to forage and returned them back into their stables at night.

Where they were free to roam under guidance?

Line 239 Table 2
Stables: The elephants were not ridden; however, they were herded by the handlers during the day and locked up in individual stables at night.

Where? On the reserve or in a boma?

Boma The elephants were moved to an electrically fenced 1.7 ha, open-air boma#. They were allowed to roam free within a 5 km radius of the boma during the day and at night, they were closed in together*.

#For international readers, please explain somewhere (Introduction or here) what a boma is in SA
*Back in the stables???


Line 251 Movement Data
African Wildlife Tracking GPS collars

You may need to provide more information here or in refs

Line 253 Same applies to Vectronic collar. You need to give specifications, especially if a patent
Most readers will not be familiar with this

Line 297 For diurnal movement patterns, we calculated the mean distance travelled per second for each period of the day

Is per second standard? Why not per minute? (Just for clarification). Seconds for such a large slow moving animals sounds very short

Line 321 sizes during Phases 1 and 2.

In Table 3 you have Part 1 and Part 2, but in Fig 3 and the text you mention Phase 1 and 2, which I think are the same? Change in the Table


Line 337 Furthermore, the northern, eastern, southern, and western sections of the reserve were utilized during this Phase.

Not sure what this means, as they also investigated north south, east and west during phase 1, so what is DIFFERENT in phase 2 ??
What do you mean by furthermore?


Line 350 … of the reserve when they were just released (2015, wet season, 2016, dry season).

Don’t understand what you are explaining here. Does this mean during phase 1 (wet and dry seasons), as 2016 dry is not “just after release”?

Line 357. During Phase 2, the elephants travelled through areas bordering one of the southern fence lines during wet, autumn, dry and spring seasons.

This is not shown in Fig 5

Lines 371 -373 Phases 1 and 2, the elephants travelled at significantly (p< 0.0001) faster speeds during the day (12:00 -14:00) and at dusk (18:00 - 20:00) than at dawn (05:00 - 07:00) and at night (22:00 – 02:00) The elephants also travelled at significantly faster speeds during Dawn than Night

Upper or lower case? Please check for consistency

Line 390 Figure 7 SPGR elephants’ average hourly displacements (+SD) 360 days post release (a) (Phase 1) compared to when free roaming on SPGR for 57 months (b) (Phase 2).

A), a), or (a) ??? Please check for consistency

Line 508 Additionally, the elephants' unfamiliarity with their new environment during Phase 1 could have overshadowed any of these environmental factors.

Would the SPGR elephants not have been familiar with the environment from the EBS rides and walking with the guides between water sources (Fig 1)?

Line 526 (Douglas-Hamilton, Krink & Vollrath, 2005; Leggett, 2006; Kikoti, 2009)

Were these fenced areas in the references?? One a corridor (possibly), the other the Kunene region, is it fenced ???

Line 538 -541 Thereafter, the elephants expanded their home range (5964-hectare) during the wet season where they utilised most parts of the reserve (Fig. 4). During Phase 2, the elephants continued to shift their focus towards different food and water sources across the different seasons (Fig. 4b). The elephants also had the largest home range size during wet season of Phase 2 (6054-hectare),

Is it a name “wet season” (dry season, spring, autumn) or in general the wet season. Check for consistency.

Line 546 Prior to their release (during the boma phase), the elephants were guided towards four water sources….

Perhaps a “pers comm” or “own observation”, or where does this info come from?

Line 556 -560 . During the dry season, the density of the water sources is less, therefore they are forced to move longer distances to try and locate new water sources, which would allow them to settle once they have found adequate sources. However, if they don’t locate such sources, they need to return to sources they know or increase their distance of movement to search for other sources.

Are you especially using the present tense as it still applies today??? Or hypothetically? Otherwise use past tense as it happened a few years ago

Line 569 …. due to elephants’ ability to remember the spatial location of other elephants
Or trying to avoid competition? Or becoming securer with more elephants present?

Line 527 … displacement was investigated (Figs. 10 & 11)

What are the journal requirements, Fig. or Figs.?

Line 600 strategies to meet their physiological needs (thermoregulation), which they did not necessarily require during their time in captivity.

Do you have a reference or idea explaining why this may be different to captivity (where they could also move around during the day).

Line 612 This herd preferred to cross the open pans to water during dusk and night-time..

Perhaps somewhere an indication that KKR is much hotter with less vegetation cover for shade than SPRG????

Line 644 Both reintegrated herds showed the least movement at midday and increased movement during dusk and dawn,

Not sure this is entirely correct as the SPGR elephants moved faster, but displaced less over midday ? Or does movement refer ONLY to displacement?

Lines 657 When we reflect on what could have contributed towards the resilience of these reintegrated elephants, we found that appropriate release date (season),

Either past of present tense for reflect/reflected and find/found

Acknowledgements
Line 668 Furthermore, we would like to acknowledge TJ Steyn, owner of Shambala Private Nature Reserve (SPGR), and !Khamab Kalahari Reserve (KKR) for granting us access to the collar data of their elephant herds.

Either a comma, or “as well as” otherwise its reads as if TJ was owner of both
Appendix
Table A2 Animal Species KKR
The reserves also have number of large herbivores such as African buffalo, gemsbok
More than one reserve???

Line 821 References
Mole MA, DAraujo…. Is this correct, a capital A??

Reviewer 2 ·

Basic reporting

The English used lacks clarity in many places, such that the information presented can be difficult to understand. There are several instances of repetition and the authors have placed content in the wrong sections on multiple occasions. Figures and tables could be greatly simplified to aid interpretation.

Experimental design

The research question could fill a meaningful gap but the analysis and presentation of findings is lacking. Methods are lacking crucial information, such as the fix schedule of GPS collars, how literature was identified to document wild elephant movement. Explanations for definitions of seasons and time periods are lacking. In the results, critical statistical information is missing, together with sample sizes. Methods and results follow a different order and some information appears in the discussion that was not presented in the methods or results. There is mention of elephant sedation/anaesthetic but no description of the process, drugs, doses involved.

Validity of the findings

Given the lack of information in methods, it is difficult to determine whether the results are completely valid. Much of the results contain text relating to visual interpretation of figures and maps, when other statistics such as HR overlap could have been used to support the validity of the results.
There is no recognition that collaring one animal in a herd may not be representative of the entire herd’s movements, particularly in the context of an artificial herd put together by humans that contains more than one related group and some adolescent males. Overall, the interpretation of the findings tends to exaggerate without identifying potential confounding factors.

Additional comments

With a lot of re-working, this could be a valuable ms to provide information and recommendations on elephant releases that could be used for similar future endeavours. More detailed comments re below.

L38-9. This is difficult to assert with a sample size of two herds in two different environments. Such differences may not be herd-specific, rather a product of the process. The same herd released in a different environment may not behave in the same way. The authors suggest this, so the term “herd-specific” is the issue here
L52-3. What determined the inclusion of animals in Table A1? L 79-81 refer to several other releases but these do not appear in Table A1. Why not?
L54. Which studies? They should be referenced here
L57-9. Where are the citations for this statement?
L60. Please check the consistency of your referencing style. Here you include author initials and use “and”. Elsewhere no initials and “&”
L60. What do you mean by “dire consequences”?
L60. Unclear which factors are being referred to.
L61. What happens if an animal is unmanageable?
L64. More cost-effective and replicable than what?
L68. This sounds as if the ultimate aim is to reduce numbers in captivity. Is that the aim, or is the aim to increase welfare for individuals by releasing them?
L73. Check your referencing style. Here repeated
L75. Which doubts? What are those doubts founded on?
L82. Which aspect of dung relates to health?
L85. Long-term success metrics have not been obviously defined previously so it is unclear what this statement is referring to.
L87-9. Where are the references for this statement?
L89. What is meant by “balanced herd dynamics”?
L90-5. So in this case does reintegration success just mean survival and reproduction?
L98. Is reintroduction the same as reintegration? I would suggest using consistent language, as the definition of similar terms can vary.
L102. What do you mean by “unbalanced bull hierarchies”?
L104. Who were these encounters fatal to? Elephants and rhinos?
L111-4. Where are the references for this statement?
L114. Which parameters are being referred to here?
L131. Remove “as expected”. This would suggest that you are testing a hypothesis rather than reviewing existing literature
L132-3. Surely this would depend on the distribution of waterholes?
L142. Which country is Sioma in?
L143. Remove “It is well known that”
L146. “A riverine” is not a common habitat type. Riverine forest perhaps?
L148-9. At which scale? Presumably other elephants are also moving so remembering one location may not be very useful. Or does this refer to a elephants within a herd and their relative positions?
L152-4. The wording here is clumsy and should be rephrased
L152-60. Could this be summarized more succinctly? Otherwise this is a full paragraph dedicated to just one study
L171-3. This has already been stated and does not need to be repeated
L177. What exactly is meant by “an adaptability”?
L184. What about holding orphaned young animals, as happens in East Africa and Botswana?
Table 1. This table holds far too much information in a confusing and inconsistent format. Times when elephants were used for safaris (e.g. 08:00-10:00) are not relevant. Information on deceased animals is not relevant. There should be more rows with less information in each. E.g. Number of adult females. Number of adult males. Country of origin. Location of captivity. Location of release. Duration of captivity. Etc. There is no need to repeat information in Table 1 that is presented in Fig. 2, such as births and translocations. Fig. 2 presents a much clearer view of this than the description in Table 1.
L220. Individual herds
L224-5. This sentence is redundant as the fact that the processes were different has already been stated.
L225-6. Why did this process take a year?
L233. Be consistent with the date format
L235. There are descriptions of the early phases in the text and then nothing about what happened after the herd was translocated. You need to be consistent: explain everything in the text and table or just in text or just in table.
Table 2. For the SPGR, dates are not overlapping, whereas they are for KKR. Be consistent in the reporting. KKR post-release stage 4. How did this stage last until the same day as stage 5 if there were problems and the herd was brought back to the boma stage? Stage 5 should then show how long they were in the boma stage for and Stage 6 would be the final release.
L243-5. How can you be sure that one elephant was representative of the entire herd? Not all animals were related and there were some teenage bulls that may have left the herd
L248-9. Repetition of previous statement, remove
L249-51. What was the GPS fix schedule? Did the collars transmit GPS data via satellite or GSM or were they store-on-board?
L252-3. This is not clearly explained. Why was a new collar needed because of new elephants arriving? Did the old collar stop working? That would be a better explanation
L253-4. Please explain the immobilization procedure – where was the animal darted from, by whom, with which drugs and doses, how long did the process take
L256-7. Does this mean that this was when the collars were fitted?
L259. Please be consistent in time period. Month 0 – 12 and month 57-69
L259. You have already talked about phases for the release of elephants, so to define phases in a different way here is confusing. Please use a different term, or at least number them in a manner consistent with the release phases and define them in the same table.
L261. Why was this 4 year period selected?
L270-71. Why are the seasons of different lengths? How exactly were they defined?
L274. What was the temporal resolution of the location data?
L276-7. Why was this error value selected?
L278. How did you calculate the UDs? Which function, which parameters?
L279-81. Please put information on movement paths together and information on UDs together, without jumping around
L289. How did you find comparable information for wild elephants in the literature? What was your process for finding relevant literature and how did you extract information? How many publications/studies did you consider?
L290. Did the literature you found include the same diurnal periods?
L291-2. No. This allowed you to see if, 4 years after release, reintegrated herds approximated wild herd movements. You would not be able to work out when this approximation happened
L293. Does this mean total movement path length? Total distance travelled from the release site?
L296. Why such fine scale resolution? Without knowing the fix schedule, this is difficult to interpret but movement per second seems to be too fine a scale and would require substantial extrapolation if GPS coordinates were not recorded at least every minute
L296-7. How did you define day periods? This has not been explained
L298. The total duration only adds up to 10 hours. What about the rest of the 24-hour period?
Results section need to follow the same order and the methods section. HR was last in methods, so should not be first in results. Every section needs to present information about sample sizes and mean ± SD values for different parameters. These could take the form of a summary table.
L313. The results should begin with sample size. How many GPS fixes were collected per animal? Were there any failures in the collars?
L315-7. The second sentence should be first and the first one deleted, replaced with (Fig.1) at the end of the first sentence. You should only refer to tables and figures indirectly in brackets.
HR section. What is the difference between HR and UD? These terms need to be defined. Where are the statistical results for this section? Why bother running statistics if you don’t report them and just show figures?
L319-20. Same comment as above about how to refer to a figure. This needs to be changed throughout
Fig. 3. Why does it matter that a wild herd was translocated in 2020? The caption is repetitive
UD section. You don’t seem to have made any calculations relating to water or food resources and these were not mentioned in the methods so you cannot draw conclusions about those. Likewise, this is first mention of reserves being divided into cardinal sections, so those results are meaningless. There is no explanation in the methods of what UD generation is supposed to show so these results are difficult to interpret and appear to be largely based around anecdotal descriptions of the area covered by UDs. It would make more sense to look at overlap of UDs in different seasons/phases and generate some reliable results.
Figs 4 & 5. The legends should be simplified to a gradient showing high and low use
L371. Be consistent. Why are Dawn and Night capitalized here but nowhere else?
Movement metrics section. It is unclear why some results are reported in m/s and others in m/h, nor why day periods are shown in Fig.6 and 8 whereas all hours are showing in Fig. 7 and 9. Only day periods are mentioned in the methods section so Figs 7 & 9 should be removed. Overall, this entire section could be simplified by stating e.g. that neither herd showed any significant difference in mean travelling speed per phase (SPGR result, KKR result). If there were no differences, data from both phases could be combined for this and displayed in one graph per herd, reducing the number of redundant graphs. Graphs are really only needed to highlight differences.
Daily displacement section. Same comments about clarity. These results could be presented much more succinctly and clearly by combining information from both herds when findings are the same and highlighting differences.
Figs. 10 & 11. What do outlier groups 1 & 2 refer to?
L474-5. There was no such comparison in the results
L476-7. This seems like a limitation rather than a strength. If the herds, release protocols and habitats are so different, it is impossible to determine what is causing the variation in your results
L480-2. Unnecessary repetition
L483-4. No need to refer back to figures, these have already been presented in the results
L485-6. This is difficult to ascertain as the reason for changes is difficult to work out
L489-96. Almost exact repetition of information from introduction. This needs to be reworded
L496-8. This information belongs in the previous paragraph
L495-6. Neither revisitation rates nor path density were considered so this sentence is irrelevant
L524. Hectare
L513-521. This information belongs in the results. There should be clear explanations of how information from literature was sourced in the methods and a summary of findings from literature presented in the results. The discussion should be reserved for interpretation of results. So here you should simply be saying that HR sizes were consistent with those for fenced elephants published in existing literature.
L530-2. Methods stated that data collection only began after elephants were released. So this information needs to be presented sooner and, if being considered as part of the data, the statement about when data were collected needs to be changed.
L532-4. Another assumption of a positive outcome. Are there any data to show that vegetation changes or variation in habitat distribution did not mean that elephants needed to leave that area?
L534-40. This information belongs in results for HR sizes. Not discussion about UDs
L538-41. What does this mean? Reporting that there is a seasonal difference is not enough in the discussion. Why is this important?
L543-7. Again, this information needs to be presented sooner, not a surprise in the discussion
L554-8. This information is poorly presented, needs to be rephrased
L561. There is no information presented in the results to support this assertion. Time spent feeding was not assessed
L564-6. This explanation is not clear. As previously said, elephants move so remembering their locations does not make sense
L569-70. Redundant sentence, remove
L580-3. If thermoregulation is the main driver, why try to explain movements with leaf sprouting/falling?
L582-3. Why is this information relevant here?
L587-95. This information was presented in the results and should not be repeated here
L607-14. Already presented in the results
L621-2. Why talk about poaching to then rule it out?
Table A1. There are two number 1s

Reviewer 3 ·

Basic reporting

The language and style is generally clear and suitable, but there are some places where more technical or scientifically appropriate terminology could be used. I have detailed these in the specific comments below.

The literature and references are generally sufficient, although I think more information or background literature is required in place, as detailed below. More importantly, I think the Introduction and analyses sections require substantial re-organisation. The Introduction contains a lot of useful information, but it does not flow well currently. I suggest restructuring it quite substantially, as explained below.

Thank you for sharing the raw data, and code, but I do think the analysis and Results sections will similarly benefit from some restructuring. I suggest aiming to match the order of the aims, and be clear about what research questions are being asked to meet each aim. This will help readers to easily understand what data was analysed, where, and how. This ordering can then be repeated in the Results section, to again allow readers to easily follow the data treatment and argument.

The study results are relevant, and interesting to document, but I do think that one of the stated aims – to compare the movement patterns of these reintegrated elephants to that of wild elephants – is not very effective. I will say more about the details of this comparison below, but essentially I think that given a direct, statistical comparison is not possible, this does not work well as an overall aim. I suggest replacing this aim with a simple statement that the study aimed to document the movement patterns of reintegrated elephants over time, within fenced wildlife reserves. That is something this paper can – and does – achieve.

Experimental design

This original, primary research of an observational study of elephants fits within the Aims and Scope of the journal. It is stated how the research fills an identified knowledge gap, but I think the aims and research questions could be better defined, to fit with the kinds of analysis that can/are actually conducted here, so that conjecture can be limited. I have made suggested edits to the aims below.

I am happy with the ethical statement provided, and agree that suitable ethical standards were met as part of this data collection. The Methods are largely well described, although some additional information seems critical to me, to help in interpretation of the results – as I outline below. The tracking collar data provides a rigorous data set, and there are clear attempts to analyse this data in a suitable way. However, I am a bit concerned that some analyses are not suitable, as detailed below, and would benefit from additional work.

Validity of the findings

I have a serious concern currently that we do not know anything about which elephant provided the movement data for each herd and, therefore, we do not know if these movements represent the ranging of an individual who spent a lot of time alone, or if they really represent the movements of the whole herd. In wild systems, male and female elephants do not typically range together for long periods. We need some sort of metric that illustrates how representative this data is of the movements of the whole herd, or if we are looking at the data for a lone animal. Even simply giving a % of time that the collared individual was seen with 1, 2, or all the other adults in the herd, in phases 1 and 2, would enable much greater insight into the data.

I am also not convinced by the choice of statistical test in parts, as detailed below.

These two things currently make it difficult to comment fully on the validity of the findings, but I do think there is interesting data here that is worth documenting.

Additional comments

This is an interesting and important study documenting the movement patterns of ‘reintegrated’ elephants, released from captive settings to fenced wildlife reserves. I think this is very important information to record and report and I would certainly like to see it published, but I do feel the manuscript and analysis need work before it is suitable for publication. I shall detail my specific comments in the order they appear in the manuscript.

Abstract
Line 22: Is the term ‘reintegration’ really only used for captive elephants, or should this read as ‘refers to the process of rehabilitating captive animals…’, before then separately making clear that this study is about reintegrating elephants in particular?
Lines 30-34: I think some of these stated aims need to be re-worked, to better fit with the data analysis conducted and conclusions that can be drawn. I suggest instead phrasing as aiming to (1) document and describe the movement patterns of two herds of reintegrated elephants; and (2) investigate any changes in ranging patterns between the first and fourth year after release.

I think the currently stated aim of comparing these herds movement to wild elephants is difficult to achieve quantitatively. Given that can really only be considered qualitatively, I think it would be best to remove that as an aim, whilst retaining the discussion of this in the Discussion section. Similarly, the aim to examine if the two herds exhibited diverse responses is also not really handled statistically at the moment. I think directly comparing a lot of the metrics described in this paper would be problematic for Phase 1, given the two herds were released in different seasons - making it impossible to rule out general habituation to/familiarity with the new environment in any observed seasonal differences.

Introduction
I think the Introduction section needs to be reorganised, to flow better from the broad context to the specific aims of this paper. I suggest beginning by outlining the problem with captivity for elephants, then explaining reintegration as a possible solution for captive elephants within South Africa, before discussing what we know about reintegration of elephants so far - re-organising the material in lines 79 to 105 to flow without jumping back and forth.

Line 87-89: Do we know for sure (e.g. with published data) that the success of reintegration hinges on things like social dynamics, or is that based on practical experience? Either some sort of reference is needed here, or an acknowledgement that this is a personal observation from the rehabilitators.

The section from lines 106 to 115 could be broken into its own paragraph, I think, to make it clear that understanding movement is important to understanding reintegration success. Then move on to (briefly) explain the South African context, and then what factors can influence movement within the typical fenced, limited reserves of SA – perhaps breaking the current long section at around line 134 – when moving from talking about water holes to electric fences. Much of the evidence you cite here is derived from this South African context, but I think it would be good to make this clear.

Finally, I think the Introduction should end with the statement of the aims (adapted, as above), outlined in the paragraph beginning on line 167.

Methods
Study animals section – I think it needs to be made clear here that all data comes from one individual animal per herd, and the sex and age of this collared animal should be made clear. To me, this information is absolutely critical for how we interpret all the movement data: are we looking at the movements of a male or female?

Table 1 could be clearer. What time point does the total number of elephants on the reserve refer to – now, or when they were released? Please give the origin of all the adults that were released, add in a row for any deaths, and make it clear how many elephants, if any, were on each reserve before these reintegrations occurred.

Table 2, KKR section 4, refers to a phase 2b, but there is nothing labelled as 2b in the table. Should it be 3b?

Movement data
This section, from line 241, needs to give more information about which elephant was collared in each herd, and also give details about the cohesiveness of the herds. I realise that early in Phase 1 of each reintegration, it is likely the whole mixed-sex herd remained close together. But I would expect that after some time, the males and females would have spent more time apart. We need to know whose data we are looking at for the movements, and have some kind of measure of how often the herd was seen together/ how often the collared animal was with others, so we can know if the movement data that is analysed is representative of just one elephant, the whole herd for the whole four years, or some mid-point between those two extremes. This is critical, in my opinion.

Movement behaviour
I think more information is needed here on how utilisation distributions were calculated (without having to look at the R code), and why the BBMM was used (again, without having to look at the explanation in the R code)

Statistical analysis
Line 287 – no (statistical) comparison with wild movements is performed, so I think this section needs to be renamed.
I suggest organising this analysis section according to the (revised) aims, making it clear what research questions are being asked to speak to each aim, and how they are being analysed – the data input, and tests used.

Results
I suggest following the same order used in the Analysis section to organise the Results, so it is clear exactly what question can be answered with each analysis/result presented. This will make all the results easier to follow and interpret, and contain data exploration to only that which is really necessary to meet the aims.

Please provide means and SD (or equivalent summary statistics) for all comparisons. It is very difficult to interpret statistical test outputs without also knowing the means of each group, sd, and (ideally) effect sizes of the positive results obtained.

Line 331: Is this investigation by eye alone, as no data is presented here? This should be made clear that it is a visual analysis only. Can no tests be conducted to confirm the changes in ranging from Phase 1 to 2?

Line 362: I am not convinced that Chi-square tests are suitable as used here to assess movement speeds. Chi-square tests are used for categorical frequency data, not comparing means.

Discussion
Paragraph beginning on line 495 is not clear. I think there needs to be more differentiation of what we see in unfenced elephants, and – more relevant here – what we see in elephants in fenced reserves with artificial water sources.

Home range size
Rather than listing some vastly different home range sizes recorded in very different contexts across savannah elephant range, I think it would be more useful to concentrate on the range use/kernel size of elephants living in fenced reserves in South Africa. How much of their maximum potential range (e.g. up to the fence barriers) do they typically use, and how does that compare to the kernel size/range use of these reintegrated herds? Ecology can explain a lot of the difference, but this seems like a slightly more valid and useful comparison to make than say, range sizes recorded in open systems in Kenya, or even Kruger.

---

## Round 0.2 · Minor Revisions

Both reviewers and I are satisfied with the changes incorporated in this new version, which addresses the noted observations. Now, to accept the manuscript it is only necessary to clarify some minor points that both reviewers pointed out directly about the manuscript.

·

Basic reporting

Movement patterns of two reintegrated African elephant (Loxodonta africana) herds: transitioning
from captivity to free-living (#93577)

Comments to second reviewing process
The authors have adequately answered all comments from the first reviewing. This version has greatly improved. The data is well presented, the text well structured. The Conclusions align with the research questions and results, pointing out the most important factors. This research is the first of its kind in showing that previously captive elephants can adapt to a new specific environment. A positive is the long time spanning this research. I am happy for it to be published after some small amendments to the text as following:

Line 60-61 …leading to an animal being deemed as unmanageable due it either killing humans or conspecifics.
Something grammatically wrong here

Line 62-63 Some facilities either sell or give away animals to other facilities or zoos if there is no longer commercial incentive.
Please rephrase, not clear.
Animals, being elephants?
…if they are no longer of commercial value??
...if there is no longer a commercial value in keeping them?

Line 60 Researchers have argued
Some researchers have suggested? Surely not all?

Line 75-79 Although reintegration is not new in southern Africa, studies (Ashraf et al., 2005; Angkavanish & Thitaram, 2012; Evans, Moore & Harris, 2013a,b; Perera et al., 2018; Goldenberg et al., 2021; Pretorius, Eggeling & Ganswindt, 2023) analysing the behavioural adaptations of elephants post-reintegration remain scarce and detractors still question the credibility and viability of this practice.
You state reintegration is not new in southern Africa, but then you reference two Asian studies. Maybe add Although reintegration is not new in southern Africa and Asia, studies…
Or: Although reintegration is not new in southern Africa, studies of both Asian and African elephants (……)
Or: Although reintegration is not new in southern Africa, studies (Ashraf et al., 2005; Angkavanish & Thitaram, 2012; Evans, Moore & Harris, 2013a,b; Perera et al., 2018; Goldenberg et al., 2021; Pretorius, Eggeling & Ganswindt, 2023) analysing the behavioural adaptations of both African and Asian elephants post-reintegration remain scarce…..

Line 83: National Norms and Standards for the Management of Elephants in South Africa
As this is a title, perhaps place it in inverted commas?
Allow or allows?

Line 119-121: contrary, Evans et al. (2013b) demonstrated that reintegration can indeed be successful for individual elephants after monitoring a reintegrated African elephant cow for over five years post-release (Evans, et al. 2013b).
I don’t think you need to reference the paper twice


Line 141-146: To better understand and evaluate the long-term impact of reintegration on the ranging behaviour of formerly captive elephants, it is vital to examine the natural movement patterns of wild elephants. This understanding is necessary not only for improving management and scientific knowledge but also as a benchmark for the expected behaviour of reintegrated elephants. The movement patterns of elephants can offer unique insights into their behaviour at an ecological level.
Now that you have removed the comparison to wild elephants in MS, I don't think this paragraph is relevant any longer.

Line 156: …reintroducing various wildlife species such as the elephant (Duffy et al., 2002).
reintroducing various wildlife species including the elephant (Duffy et al., 2002).

Line 158: These unnatural behavioural occurrences is are often a consequence of limited space

Line 161-162: the introduction of cull orphans originally translated from KNP
This the first mention of KNP, so write out full name
….. as well as conflicts with other species within the fenced system

Line 163: Artificial boundaries could have a confounding effect on elephants.
Why??? What do you mean by this??
Localised movement patterns and reduction of food for other species, is not a confounding effect on the elephants.


Movement data

Line 284: and through daily observation sessions conducted by the handlers

Line 301-302: All requirements as per the necessary regulations, TOPS legislation / Veterinary Regulations and the EMP were adhered for this procedure.
And N&S???
You need to give full title of TOPS and EMP, and both references ( i.e. websites) in reference section


Line 333-334: ….to 23 m for all locations (Purdon & Van Aarde, 2017). The resolution of
334 the square raster cells was set at 30m,
no gap

Line 350-352: Four periods of the day were chosen to replicate what previous studies have chosen and comprised of dawn (05:00-07:00), midday (12:00-14:00), dawn/dusk (18:00-20:00) and night (22:00-352 02:00)

Discussion

Line 585: … and individuals. initial movements patterns…
Delete s


Line 596-7: due to their decreased dependency on water during this season

Line 621-623: Addo National Park is considered a large-fenced system and researchers have shown 100% mean range size for females is 5500-hectare (reserve size = 70000-hectare), which amounts to 8% of the reserve being utilized (Whitehouse & Schoeman, 2003).
I find this confusing, if you say 100% mean range size, but then state they only used 8% of the available area. I feel this needs clarification. Maybe leave out the 100%?


Line 636 -638: When the SPGR elephants were still in captivity, they roamed in the areas between water sources 5, 6, 7, 8 and 11 under the guidance of their handlers.
I don’t see how this relates to the previous sentence, or the following one. Why do you mention this? Is there a comparison the new UD? If not, rather leave it out as the next sentence refers to the small area explored being positive ( Another positive observation….) Or place it somewhere more appropriate. Possibly after this sentence, but then discuss the difference/similarity ( why you mention it, as you did with KKR).

Line 654-658: During the dry season, the density of water sources were less, forcing them to travel longer distances to locate new water sources and settle once they have found adequate sources. However, if they don’t locate such sources, they either need to return to sources they know or increase their distance of movement to search for other sources.
I find the usage of present and past tense confusing. Either make it all past or if still applicable all present tense, and maybe mention it is still like that. As your observations and all other text are in the past, I prefer past, things may have changed since then.

Are there no artificial water sources on KKR??


Line 663-668:
Did you see them walking together as one herd?? If so mention it. If not then separate the sentences:
E.g.
The translocation of a wild herd in September 2020, could also have contributed towards the expansion into these novel areas. Due to elephants’ ability to remember the spatial location of other elephants (Bates et al., 2008), they may possibly want to either avoid competition (Dunbar, 1992) or move together as one herd for increased security (Hamilton, 1971) demonstrating social learning (Lee & Moss, 1999) and fusion patterns (Goldenberg et al., 2022).

Line 672-673: The more prominent peaks in movement during Phase 2 could indicate…..
For which reserve???



References

In the Intro you mention DEAT, maybe also need to mention if its NEMBA or N&S.
In the references I think you should also give the website for the N&S 2008, or the new one (not sure if it’s been signed off?).

In the text references you sometimes cite Name et al … and sometimes you write all names. Please amend according to specs for Journal


Table 2 Characteristics of the reserves
Animal Species KKR. You forgot to write the scientific name after African buffalo

Experimental design

No comment

Validity of the findings

no comment

Additional comments

no comment

Reviewer 3 ·

Basic reporting

The reporting is satisfactory, and I appreciate the work the authors have put in to develop the points I raised here in my original review: I think suitable background and context is now provided, and the structure of the article has been improved (although there are still a lot of tables and figures - not all of these may be necessary). The key improvement is that the results are now relevant to the aims as presented.

Experimental design

The aims and research questions are now much more suitable and better-defined, and the analysis and results are more appropriate to these aims.

Validity of the findings

As mentioned, the validity of the findings are greatly improved as they are now better presented, relevant to the aims, and suitably analysed. This has strengthened the conclusions drawn.

Additional comments

I am satisfied that the authors have adequately addressed my concerns and points in my original review. I have a few remaining points from the updated manuscript, but these will be simple to implement.

Line 67 – the definition of captive is only applicable to the South African context. To make this suitable for the wider, international readership of PeerJ, I would make this definition not so country-specific.

Line 81 – do you mean including reintegration, here, instead of ‘than reintegration’? If not, I am not sure this sentence really fits/makes sense here.

Line 203: Replace ‘proven’ with demonstrated.

Line 223-240: I am not sure how this is relevant or fits with the rationale being established here – I suggest removing this section.

Paragraph beginning on Line 246: A lot of this information does not seem to be only related to the South African context – I would say several of these factors influence movement across all open savannah elephant systems. Perhaps save the ‘south African context’ bit for the section beginning on line 262? And split this very long paragraph there.

Line 285: I don’t think this phrasing is quite correct, as Bates et al 2008 did not show any long-term recall about the locations of other elephants, just that they could keep track of other elephants in the short-term, in relation to themselves. Safer phrasing here would be ‘Tracking the locations of other elephants within a certain area could explain…”

Line 289: Given that Evans et al 2013 showed the reintegrated elephants’ movement did not differ significantly from that of wild elephants, I think the phrasing here is a bit odd. How about: ‘Evans et al (2013) assessed if the movement patterns of reintegrated elephants differed from that of their wild counterparts.’

Line 425: Figure 2 doesn’t actually show the observations that the elephants stick together, so I would remove this reference to the figure here.

Line 460: I do understand what you are getting at, but I think invoking a lack of social learning here is a bit confusing – you could just as easily argue that they have learnt (socially) to stick together. Simpler/less ambiguous phrasing would be: ‘however, it is likely this is reflection of the strong bonds formed during their captivity’.

Line 470-471: So what was the GPS fix schedule? It would be useful to state here how often data points were being collected.

Line 536: Error with mentioning dawn twice – assume the second one should be dusk , as the time is 18-20h?

Line 1042 onwards: Please give degrees of freedom and be clear about statistical test you are reporting here.

Line 1139: To remind readers, I would add the phrase ‘Visual inspection of the UD suggests…’.

---

## Round 0.3 · accepted · Accept

After carefully reviewing this latest version of your writing, I have checked the incorporation of the latest comments from the two reviewers, so I consider that the manuscript is ready for publication.